# Real-Time Hand Gesture Recognition Using Fine-Tuned Convolutional Neural Network

**DOI:** 10.3390/s22030706

**Published:** 2022-01-18

**Authors:** Jaya Prakash Sahoo, Allam Jaya Prakash, Paweł Pławiak, Saunak Samantray

**Affiliations:** 1Department of Electronics and Communication Engineering, National Institute of Technology, Rourkela 769008, Odisha, India; sahoo.jprakash@gmail.com (J.P.S.); allamjayaprakash@gmail.com (A.J.P.); 2Department of Computer Science, Faculty of Computer Science and Telecommunications, Cracow University of Technology, Warszawska 24, 31-155 Krakow, Poland; 3Institute of Theoretical and Applied Informatics, Polish Academy of Sciences, Bałtycka 5, 44-100 Gliwice, Poland; 4Department of Electronics and Tele Communication Engineering, IIIT Bhubaneswar, Bhubaneswar 751003, Odisha, India; saunaks64@gmail.com

**Keywords:** ASL, fine-tunning, hand gesture recognition, pre-trained CNN, real-time gesture recognition, score fusion

## Abstract

Hand gesture recognition is one of the most effective modes of interaction between humans and computers due to being highly flexible and user-friendly. A real-time hand gesture recognition system should aim to develop a user-independent interface with high recognition performance. Nowadays, convolutional neural networks (CNNs) show high recognition rates in image classification problems. Due to the unavailability of large labeled image samples in static hand gesture images, it is a challenging task to train deep CNN networks such as AlexNet, VGG-16 and ResNet from scratch. Therefore, inspired by CNN performance, an end-to-end fine-tuning method of a pre-trained CNN model with score-level fusion technique is proposed here to recognize hand gestures in a dataset with a low number of gesture images. The effectiveness of the proposed technique is evaluated using leave-one-subject-out cross-validation (LOO CV) and regular CV tests on two benchmark datasets. A real-time American sign language (ASL) recognition system is developed and tested using the proposed technique.

## 1. Introduction

The interface between humans and machines involves the most common modes of communication such as speech and hand gestures [1]. These types of interactions are intuitive and user-friendly. In general, people have used remote controls and joysticks as controlling devices for many human–machine interfaces. However, to operate those devices, a trained user is needed [2]. On the other hand, a hand-gesture-based interface provides higher flexibility while also being user-friendly [3] because the user has to operate a machine using only his hand in front of the camera. Several applications that use static hand gesture recognition systems are sign language interpretation [4,5], automatic television control [6], smart home interactive control [2], gaming [7], control of a software interface [8] and control of virtual environments [8]. In real-time application, high accuracy and robustness of background interference are required for the design of an efficient gesture recognition system. Therefore, the precision of the hand gesture recognition (HGR) system still provides several challenges to researchers.

The general steps for vision-based static hand gesture recognition are data acquisition, segmentation of the hand region, feature extraction and gesture classification based on identified features [7,9]. The data acquisition process for vision-based hand gesture recognition is accomplished using sensors such as web cameras [10] and the Kinect depth sensor [11,12]. The human skin-color-based technique [5,13] is used for localization of the hand from the color images, assuming the hand region contains the majority area of the image frame. In such cases, the segmentation of the hand is difficult if the hand is surrounded by the human face or body and the background color is similar to human skin color [5]. The depth threshold technique is applied on the depth image of a Kinect sensor to segment the hand region from the background [7,14]. In these cases, the hand is assumed as the closest object in front of the Kinect sensor [9]. Again, the depth image is free from background variations and human noise [15]. After that, several feature extraction techniques are applied on the segmented hand to extract the semantic information for the input gesture image. Then, the gestures are recognized using different classifiers.

In the literature, many researchers have applied hand-crafted feature extraction techniques such as shape descriptors, spatiotemporal features [16], and the recognition of hand gestures. However, these features have performed well in a specific environment while performance has degraded in varied conditions of the dataset [17]. Nowadays, deep learning techniques are used to overcome the above limitations. In these cases, convolutional neural network [17] and stacked denoising autoencoder [18] architectures are used. However, it is a challenging task to train the CNN from scratch due to following reasons [19]: (1) A huge number of level image datasets are required to train the CNN effectively. (2) High memory resources are required to train the CNN, otherwise the training remains slow. (3) Sometimes the training of the CNN also suffers from convergence issues, which requires repetitive adjustment in CNN layers and learning of hyperparameters. Therefore, the development of a CNN-based model is very tedious and time-consuming. To overcome the above issue, the dataset having less image samples adapted a transfer learning technique. In this technique, the pre-trained CNN models such as AlexNet [20], VGG [21], GoogLeNet [22] and ResNet [23] that have been trained on large label datasets are fine-tuned on the target datasets.

Therefore, an efficient and accurate hand gesture recognition model is highly essential for the recognition of hand gestures in real-time applications. To develop such a recognition model, a score-level fusion technique between two fine-tuned CNNs such as AlexNet [20] and VGG-16 [21] is proposed in this work. The contributions in this work are as follows:An end-to-end fine-tuning of the deep CNNs such as AlexNet and VGG-16 is performed on the training gesture samples of the target dataset. Then, the score-level fusion technique is applied between the output scores of the fine-tuned deep CNNs.The performance of recognition accuracy is evaluated on two publicly available benchmark American Sign Language (ASL) large-gesture class datasets.A real-time gesture recognition system is developed using the proposed technique and tested in subject-independent mode.

The rest of the paper is organized as follows. In Section 2, recent works on hand gesture recognition techniques are reviewed. The methodology of the proposed work on pre-trained CNNs is discussed in Section 3. Section 4 demonstrates the standard dataset and validation techniques used to evaluate the performance of the proposed technique. The detailed experimental results and analysis are presented in Section 5, whereas real-time implementation of the proposed technique is presented in Section 6. Finally, the paper is concluded in Section 7.

## 2. Related Works

In this section, a detailed literature survey of recent techniques for vision-based hand gesture recognition is presented. The study includes the recognition of hand gestures based on RGB cameras and depth sensors using machine learning and deep learning techniques.

### 2.1. Hand Gesture Recognition Using RGB Sensor Input

Several techniques have been proposed to recognize hand gestures in vision-based environments. Some researchers have used hand-crafted features followed by classification, and some used convolutional neural network techniques which perform both feature extraction and classification in a single network. The recent survey on these techniques is presented below. A feature-fusion-based CNN network was proposed by Chevtchenko et al. [24], where four types of features viz Hu moments, Zernike moments (ZM), Gabor filter and contour features are extracted from the input hand gesture images. The above four features with CNN structure are fused at the last fully connected layers to obtain a final gesture class. A novel technique of wristband-based contour features (WBCFs) was proposed by Lee et al. [25] for the recognition of static gestures in complex environments. A pair of black wristbands are used for both hands to segment the hand region accurately. A matching algorithm is used to recognize the gesture class. The system fails to segment the hand region accurately where the background color is back. In another work by Chevtchenko et al. [26], a combination of features and dimensions are optimized using a multi-objective genetic algorithm for static hand gesture recognition. A recognition accuracy up to 97.63% was achieved on 36 gesture poses of an MU dataset. The above accuracy was obtained with the combined features of Gaubor filter (GB) and Zernike moments (ZM) with a holdout CV test. Furthermore, a set of geometric features (SoGF) [27] such as angle, distance and curvature features, defined as local descriptors, was obtained from the contour of the hand gesture. Afterward, local descriptors were optimized using the Fisher vector, and the gestures were recognized using the support vector machine (SVM) classifier. Again, deep features [28] were extracted from the fully connected layer of AlexNet and VGG 16 for the recognition of sign language. The extracted features were classified using the SVM classifier. The recognition performance was found to be 70% using the leave-one-subject-out cross-validation (LOO CV) test on a standard dataset. The above study shows that for an RGB input image, recognition accuracy is mainly limited by variation in backgrounds, human noise and high inter-class similarity in ASL gesture poses [28].

### 2.2. Hand Gesture Recognition Using RGB-D Sensor Input

The limitations of the hand region segmentation problem can be solved using RGB-D sensors. Some of the literature on hand gesture recognition using RGB-D sensors is as follows. A novel feature descriptor, depth projection maps-based bag of contour fragments (DPM-BCF), was proposed by Feng et al. [9] to extract the bag of contour fragments (BCF) features from depth projection maps (DPMs). This shape feature is obtained from depth maps in three projection views: front, side and top view. The final shape feature is represented by concatenating the above three views. Since the feature is obtained from contour of the DPMs, they are noisier. The authors found that the front view projection provides major contribution for the recognition of hand gestures. A fusion of feature descriptors from the segmented hand are proposed by Sharma et al. [15] for the recognition of hand gestures. These features are geometric, local binary sub-pattern distance features and number of fingers. The combination of different features was tested on two datasets using the SVM classifier. A two-stage CNN network was proposed by Dadashzadeh et al. [29] for the recognition of hand gestures using depth images. The first stage of the network was used for hand region segmentation and the second stage was used to recognize the hand gestures. The shape features from the segmented image and the appearance features from the RGB image were fused together in the network before classification.

From the above literature survey, we conclude that the performance of the HGR system mainly depends on system accuracy and distinguishable features between the gesture classes in a dataset. Therefore, in this work, a score-level fusion technique between two fine-tuned CNNs is proposed to recognize static hand gestures in the vision-based environment.

## 3. Proposed Methodology

An overview of the proposed hand gesture recognition system is shown in Figure 1. As shown in the figure, the recognition of static hand gesture images is achieved by the following steps: data acquisition, pre-processing and recognition of hand gestures using proposed technique.

The step-wise operation details of static hand gesture recognition are as follows:

### 3.1. Data Acquisition

Several sensors are available to develop an HGR system. A comparison among the sensors with their advantages and limitations is presented in Table 1. The table shows that an HGR system developed using the data glove is more accurate and robust, but the user feels uncomfortable and their hand is restricted by wearing the glove [15]. Again for the leap motion sensor-based HGR, the tracking of the hand is completed with a high precision; however, the hand coverage area is lower. Compared to the other sensors, a vision-based sensor does not require any object to be put on the user’s hand, and the hand gestures are captured using the sensor with free hand [30]. This advantage of the sensor attracts researchers to develop the HGR system using the vision sensor. In this work, the Kinect V2 depth sensor is used to develop the proposed hand gesture recognition system as it is easier to segment the hand region more accurately from the image frame.

### 3.2. Preprocessing

The objective of this step is to segment the hand region from the hand gesture image frame and to resize it into the pre-trained CNN’s input image size. The color and depth map images are obtained from the Kinect depth camera as shown in Figure 2. Between both inputs, only the depth map image is considered for recognition of static hand gesture. Depth thresholding is used for segmentation of the hand region from the depth map. An empirically determined value of 10 cm [14] is chosen as a depth threshold value to segment the hand from the background as shown in Figure 2c. The maximum-area-based filtering technique is used to find the hand region and remove the noise section of the segmented image as shown in the bounding box form in Figure 2c. Following this, the bounding box region is cropped form the segmented image. Both pre-trained CNNs operate with three-channel input images. Therefore, the cropped hand gesture images are normalized to generate a single-channel image in a range from [0,255] using (Equation 1).
(1)$$D\left(x,\;y\right)=\;\left\{\begin{array}{c} \frac{\max(D)-D(x,y)}{\max(D)-\;\min(D)}\;\times 255\;\;\;\;\;if\;D\left(x,y\right)\neq \;0\\ 0\;\;\;\;\;\;\;\;\;\;\;\;\;\;\;\;\;\;\;\;\;\;\;\;\;\;\;\;\;\;\;\;\;\;\;\;\;\;\;\;\;\;\;\;\;\;\;\;\;\;\;\;\;\;\;\;\;\;\;\;\;if\;D(x,y)=\;0 \end{array}\right.$$
where *D* denotes the depth values in the depth map image and (x,y) are the pixel indices in the depth map. max(D) and min(D) are the maximum and minimum depth vales in the depth map. Conversion of a single channel to three channels is performed by applying a jet color map [31] on the single-channel hand cropped image. The hand segmented image is resized according to input image size of pre-trained CNN AlexNet and VGG-16. Therefore, all the images in the dataset are resized to a resolution 227×227×3 for fine-tuning of pre-trained AlexNet, and for fine-tuning of pre-trained VGG-16, the input image is resized to 224×224×3 image resolution.

### 3.3. Architecture of Pre-Trained CNNs and Fine-Tuning

The important layers of CNNs are convolution layer, pooling layer, fully connected layer and output softmax layer. The distinguishing features are extracted from the input images in the convolution layers. In this layer, the features are obtained using the convolution operation between a kernel with a square part of the input image of the same kernel size. A kernel or filter is a small rectangular matrix viz 3×3, 5×5, whose height and width are hyperparameters and the width of the kernel is same as the input image channel (i.e., for RGB image the channel is 3). The kernel slides over the input image to produce the feature map. The dimension of the feature map is reduced using the pooling layers keeping the most important information of the feature map. Different types of pooling operations are max pooling, average pooling, etc. The classification of images is performed by fully connected layers. Finally, the desired output gesture class is predicted from the output softmax layer [32].

In this work, two pre-trained CNN models, i.e., AlexNet and VGG-16, are used for the classification of hand-gesture images. AlexNet [20] is a deep CNN architecture used for classification of images. This CNN is composed of five convolution layers, three pooling layers and three fully-connected layers with approximately 60 million free parameters. The non-linear Rectified Linear Unit (ReLU) function is proposed in this CNN network. The advantage of ReLU over sigmoid is that ReLU helps the network to train faster than the sigmoid function [20]. The dropout layer used after the fully connected layer is also another proposition in this CNN architecture.

The VGG model [21] was proposed by Simonyan et al. for classification of ImageNet data. The model shows that the use of only (3×3) convolutional kernels in the network achieves significant improvement in the classification performance. In this model, the number of convolutional layers is more as compared to AlexNet. The VGG network is categorized as VGG-16 and VGG-19 based on the number of layers in the network. In this work, the pre-trained VGG-16 model is used for recognition of static hand gestures.

Fine-tuning of pre-trained CNN: Generally, during the training of CNN, the weight and bias of each convolutional layer are randomly initialized with zero mean and small standard deviation. Following this, the weights are updated according to back propagation of error. Finally, on completion of the training process, the updated weights are frozen. The training process requires a large number of image datasets for training. In the CNN, the number of weights increases with an increase in the number of layers. However, if the dataset size is small, then the CNN model may lead to an undesirable local minimum for the cost function [33]. Therefore, the alternative solution for this problem is, instead of training the CNN from scratch, the weight and bias of the convolution layers can be initialized with the weights of pre-trained CNN model [34].

The fine-tuning of the CNN starts with transferring the weights of the pre-trained CNN to a network which learns on the target dataset. In this process, the weights and biases of the pre-trained CNN model are updated after each iteration on the target dataset as shown in Figure 3. As shown in the figure, the fine-tunning process of the pre-trained AlexNet is carried out on the HUST-ASL dataset. In this process, the last fully connected layer of the pre-trained AlexNet is changed into 34 nodes, which is the number of classes in the dataset. Then, the model is fine-tuned according to the hyperparameter setting.

### 3.4. Normalization

In general, normalization of the output score is performed to decrease the score variabilities among the different models and to put both models’ score values on the same scale. Therefore, the output scores from the two fine-tuned CNNs are put into the interval [0, 1] using the min-max normalization technique [35]. The normalized score of *s* (s∈S) is denoted as $s^{\prime }$. The normalized score is calculated using (Equation 2).
(2)$$s^{\prime }=\frac{s-\min(S)}{\max(S)-\min(S)}$$
where *S* is set of raw output score vectors of *s* obtained from fine-tuned CNN model, and min(S) and max(S) are the minimum and maximum values in *S*, respectively.

### 3.5. Score-Level Fusion Technique between Two Fine-Tuned CNNs

Two different score vectors, *S*1 and *S*2, are constructed after score normalization using (Equation 2), with $S_{1}$ and $S_{2}$ corresponding to the normalized scores of two different fine-tuned CNNs. Both output scores are combined together using the sum-ruled-based fusion method [35]. The normalized score vectors of two fine-tuned CNN models ($S_{1}$, $S_{2}$) are combined together to form a single score vector using (Equation 3).
(3)$$fs=wS_{1}+\left(1-w\right)S_{2}$$

The notations S1 and S2 are the score vectors of the fine-tuned AlexNet and VGG-16 models, respectively. The optimal weight value (*w*) is assigned to the score vector of the model. This is obtained between [0, 1] using a grid-search algorithm [36]. The optimal weight value is found to be 0.5 for both datasets using the above search algorithm.

## 4. Experimental Evaluation

### 4.1. Benchmark Datasets

The effectiveness of the proposed technique is evaluated using two publicly available benchmark static hand gesture datasets. Detailed information on the datasets is provided in the following subsections.

#### 4.1.1. Massey University (MU) Dataset

The MU dataset [37] comprises of a total 2515 color ASL static hand gesture images collected by five users. This dataset consists of 36 static ASL gesture poses (10 ASL digit signs 0–9 and 26 ASL alphabet signs A–Z) with illumination variations in five different angles such as top, bottom, left, right and diffuse. The other challenges in this dataset are rotation, scale, and the hand shapes of users. Since, in this work, our objective is to recognize the ASL gesture poses, we used this dataset to test the gesture recognition performance using our proposed technique.

#### 4.1.2. HUST American Sign Language (HUST-ASL) Dataset

The HUST-ASL dataset [9] is developed using a Microsoft Kinect camera and contains a total of 5440 static hand gesture samples of color images and their corresponding depth maps. Thirty-four gesture poses of the dataset include 24 ASL alphabet sign (except ‘j’ and ‘z’) and 10 ASL digits (0 to 9). Each gesture sample in a class is repeated 16 times, and 10 subjects are used for the development of dataset. This dataset is more challenging due to complex background, human noises and some gesture samples are collected by rotating their wrist or elbows to a certain degree of rotation.

### 4.2. Data Analysis Using Validation Technique

Two cross-validation (CV) methods [9], leave-one-subject-out (LOO) CV and regular CV, are used in the experiment. The LOO CV test is a user-independent CV technique. In this technique, the trained model is developed using U−1 user gesture images, where *U* is the total number of users in the dataset, and the model is tested using the remaining users’ gesture images. This test is repeated *U* times to obtain the performance in mean accuracy. In a regular CV test, the total gesture images in a dataset are randomly divided into 50–50% as training and testing images. The model is trained using the training gesture samples and tested using the test images to obtain accuracy. This process is repeated ten times and the result is found in mean accuracy.

### 4.3. Setting of Hyperparameters for Fine-Tuning

The process for setting up the hyperparameters of a fine-tuned CNN model is as follows: The network is fine-tuned with a batch size of 32 for the datasets. The initial learning rate of the pre-trained CNN is set to a value of 0.001, and a momentum value is set as 0.9 for both the datasets. The work is developed using the Matlab deep-learning toolbox.

## 5. Results and Analysis

In this work, the experiments are performed using Intel Xeon 2.40-GHz CPU with 32 GB RAM and 8 GB NVIDIA GPU card. The performance of the proposed method is obtained in mean accuracy.

### 5.1. Performance Evaluation

The mean accuracy of test gesture samples with the score fusion between the two fine-tuned CNNs is shown in Table 2. The tabulation result shows that the proposed technique performs better in terms of mean accuracy (average ± standard deviation) compared to both fine-tuned CNNs on the above datasets. The proposed technique performs at 90.26% and 56.18% mean accuracy with the LOO CV test on the MU dataset and HUST-ASL dataset, respectively. Similarly, for the regular CV, test the performance of the proposed technique shows a mean accuracy of 98.14% and 64.55% for both datasets. The subject-wise LOO CV test recognition performance in both the datasets is represented in Figure 4. The result shows that the performance of the LOO CV test is higher compared to the regular CV test. The reason for this is described below. The LOO CV technique is a user-independent CV technique [9]. In this technique, the performance of the trained model is evaluated using the gesture samples of the user, who does not take part in the model development. However, in regular CV, the gesture samples of all the users in the dataset take part in the training and testing processes. Hence, this CV test is user-biased. Therefore, the model performance using the regular CV test is higher than the LOO CV test. The confusion matrices of the test gesture samples in the MU dataset and HUST dataset using the LOO CV test are shown in Figure 5 and Figure 6, respectively. The most confusing gesture poses are ‘6’ and ‘w’ in the MU dataset. A total of 52.9% of gesture pose ‘6’ is misclassified to gesture pose ‘w’, and 48.6% of gesture pose ‘w’ is misclassified to gesture pose ’6’ as shown in Figure 5. The visualization of gesture pose similarity between poses ‘6’ and ‘w’ is shown in Figure 7a. The location of the thumb finger in the figure is confusing to distinguish with the human eye.

### 5.2. Comparison with Earlier Methods

In this section, the proposed technique is compared with state-of-art methods using the above two publicly available standard datasets.

#### 5.2.1. Comparison Results on MU Dataset

The comparison results of proposed technique with the existing techniques on this dataset are shown in Table 3. The table shows that the result obtained using proposed technique is superior than the benchmark methods in the LOO CV test. The experimental results of the CNN are obtained from [24]. In order to compare the performance of the proposed technique with the existing techniques using the holdout CV test, the experiments are conducted using the method discussed in [24]. In the holdout CV test, 80% of the gesture samples of the dataset are used to train the model, and the remaining 20% of the samples are used as test data to obtain the model performance. Table 4 shows the comparison of proposed technique on feature fusion, such as the GB-ZM and GB-HU results. The performance of the proposed technique shows better results than the GB-ZM and GB-HU results.

#### 5.2.2. Comparison Results on HUST-ASL Dataset

The performance comparison of the HUST-ASL dataset with the earlier reported techniques on the LOO CV and regular CV tests is shown in Table 5. The tabulation result shows that the proposed technique provides 5.85% and 8.01% higher mean accuracy than the earlier reported techniques on the LOO CV and regular CV tests, respectively. A fine-tuned CNN model is able to extract the feature from the hand gesture image accurately. This shows the effectiveness of fine-tuned CNNs.

### 5.3. Error Analysis on Both Datasets

The misclassification errors are analyzed from the confusion matrices of both the datasets. The gesture poses ‘6’ and ‘w’ are observed as the most misclassified class in both datasets. This misclassification is due to gesture being similar in shape as shown in Figure 7a. Therefore, the analysis of similarity in gesture poses with no fingers is visualized in Figure 7b. The figure shows that there are eight gesture poses in the dataset which are no finer but similar in shape. The comparative analysis of the above gesture poses in mean accuracy is shown Table 6. The tabulation result of the above table shows that the performance of gesture poses ‘0’, ‘o’ and ‘s’, ‘t’ provides poor mean accuracy due to their similar gesture pose.

The poor performance for these similar gesture poses is due to the following reasons: (1) The CNN network automatically extracts the features from the input gesture, which may not provide sufficiently distinguished characteristics between the gesture poses. (2) The similarity between some gesture poses is very high, such as the poses ‘m’, ‘n’, ‘s’ and ‘t’ in which the position of thumb finger is the only difference to distinguish the poses, as in Figure 7b. This may cause confusion in the extraction of distinguished features. (3) The gesture poses of the HUST-ASL dataset are collected in some angular diversity, due to which the confusion between the gesture poses increases and which may lead to misclassification.

### 5.4. Computational Time of the Proposed Method on Real-Time Data

In this subsection, the computational time (*s*) for the proposed hand gesture recognition system is evaluated with real-time hand gesture data. The pre-processing time of the proposed system for hand region segmentation with image resizing of the input depth image is 0.0969 s. The fine-tuned CNNs utilized 0.4236 s to generate individual scores, and finally, the recognition of gestures using the score fusion technique takes 0.0014 s. Thus, the total time to recognize a gesture pose using the proposed technique is 0.5219 s.

## 6. Recognition of ASL Gestures in Real Time

The recognition of ASL gesture poses in real time is implemented in Matlab with a Kinect V2 depth camera. The step-wise operation for the hand gesture recognition is shown in Figure 8. As shown in the figure, both color and depth map images are taken from the Kinect V2 depth camera. The depth map image is considered as input for the recognition of static hand gestures. The recognition of a real-time input depth image is obtained as follows:(1)*Segmentation of hand region*: In this step, the pixel values above *d* + 10 cm are marked as zero in the depth map, where *d* is the first pixel in the search space from the Kinect camera.(2)*Conversion from one to three channels*: The pixel values of a hand segmented image are normalized from [0,255] and converted into three channels using a jet color map.(3)*Image resize*: The three-channel color image is resized to 227×227×3 and 224×224×3 image resolution according to fine-tuned AlexNet and VGG-16 CNN model input image sizes, respectively.(4)*Fine-tuned CNNs*: The resized input gesture is given as input to both fine-tuned CNNs to obtain the output score. The fine-tuned CNNs are taken from the CNN model trained on the HUST-ASL dataset.(5)*Score fusion*: Both score vectors are normalized using min-max normalization, and normalized scores are combined together using (Equation 3) with the weight value *w* as 0.5.(6)*Recognized hand gesture*: The output gesture pose is the maximum value obtained in the fused score vector.

Some examples of real-time recognition of hand gesture poses are illustrated in the Figure 9. The figure shows the correctly recognized ASL gesture poses are ‘4’, ‘7’, ‘d’ and ‘i’.

## 7. Conclusions

This paper has introduced a score-level fusion technique between two fine-tunned CNNs for the recognition of vision-based static hand gestures. The proposed network eliminates the requirement of illumination variation, rotation and hand region segmentation as pre-processing steps for the color image MU dataset. Due to the depth thresholding technique, the segmentation process of the hand regions became easier even in the presence of human noise and complex backgrounds. The experimental results prove that the HGR performance using the proposed technique is superior than the earlier works on two benchmarked datasets. Moreover, the proposed technique is able to distinguish the majority of closely related gesture poses accurately, due to which the recognition performance is improved. For the HUST-ASL dataset, the LOO CV test performance is limited as the few gesture poses are collected in out-of-plane rotation. The proposed technique is also used to recognized the ASL gesture poses in real time. In future work, some specific shape-based feature extraction techniques from different views of the gesture pose may be introduced in the current HGR system to handle the out-of-plane gesture poses.

## Figures and Tables

**Figure 1 sensors-22-00706-f001:**
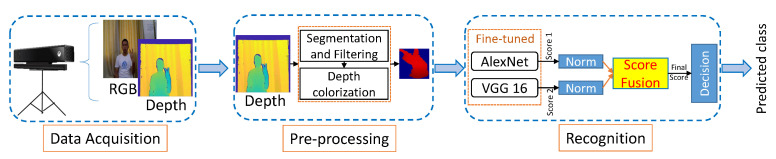
The proposed framework for the recognition of static hand gesture images.

**Figure 2 sensors-22-00706-f002:**
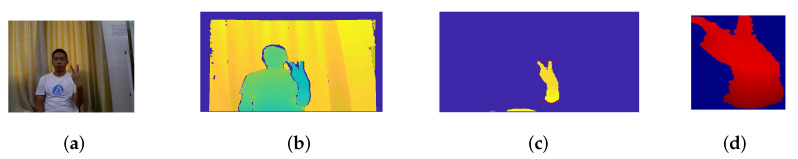
Simulation result of preprocessing step on HUST-ASL dataset: (**a**) Color image of the RGB-D images. (**b**) Depth map of the corresponding color image. (**c**) Localization of hand from the depth map using depth thresholding and removal of noise. (**d**) Resize of the hand segmented image according to pre-trained CNN input size.

**Figure 3 sensors-22-00706-f003:**
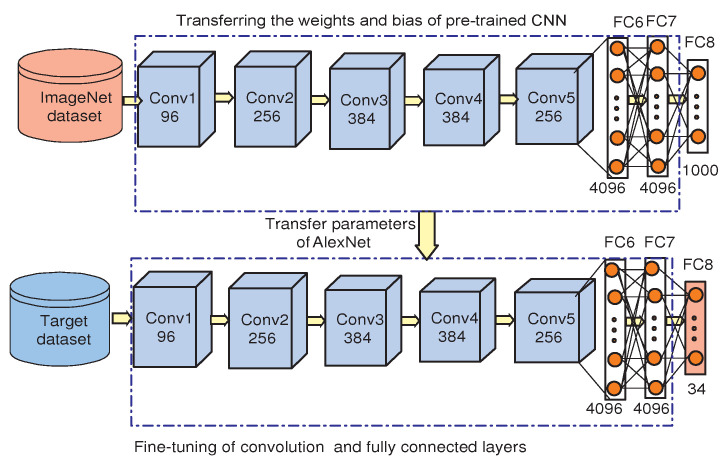
Fine-tuning process using pre-trained AlexNet on target hand gesture dataset.

**Figure 4 sensors-22-00706-f004:**
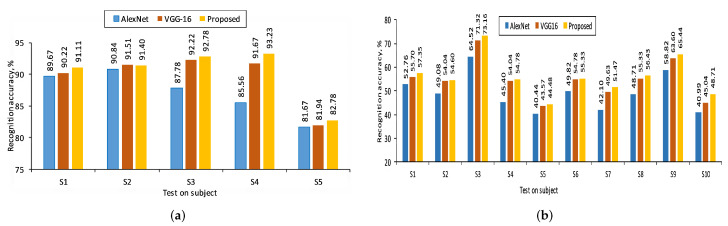
The subject-wise comparison of recognition accuracy in the LOO CV test on both datasets used: (**a**) MU dataset; (**b**) HUST ASL dataset.

**Figure 5 sensors-22-00706-f005:**
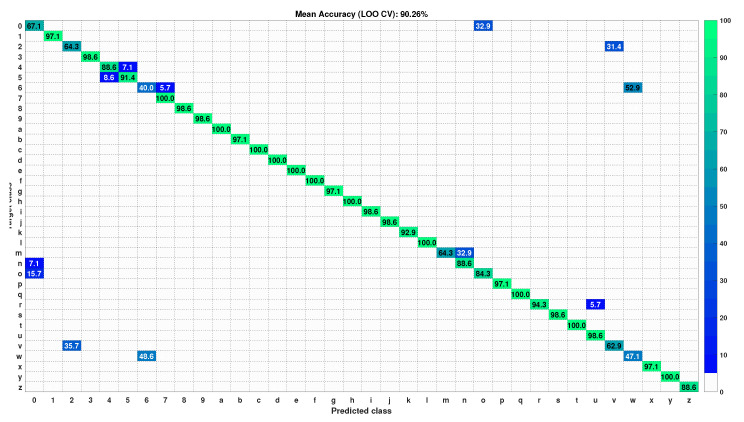
Confusion matrix of MU dataset with test gesture samples in LOO CV test.

**Figure 6 sensors-22-00706-f006:**
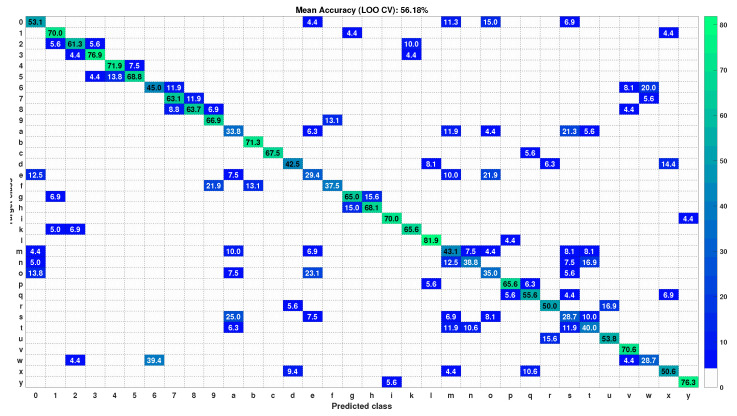
Confusion matrix of HUST dataset with test gesture samples in LOO CV test.

**Figure 7 sensors-22-00706-f007:**
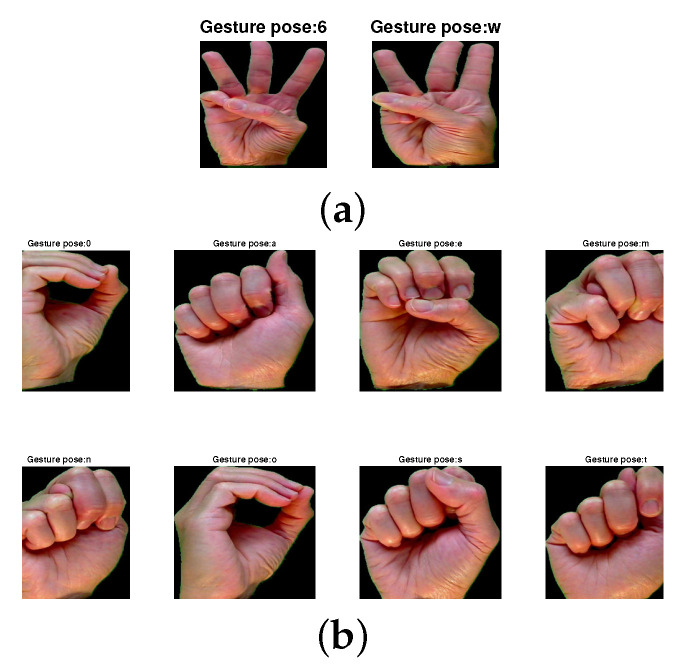
Similar gesture poses of MU dataset. (**a**) Most confused gesture poses of MU dataset ‘6’ and ‘w’. (**b**) Static ASL gesture poses with without any fingers held out are ‘0’, ‘a’, ‘e’, ‘m’, ‘n’, ‘o’, ‘s’ and ‘t’.

**Figure 8 sensors-22-00706-f008:**
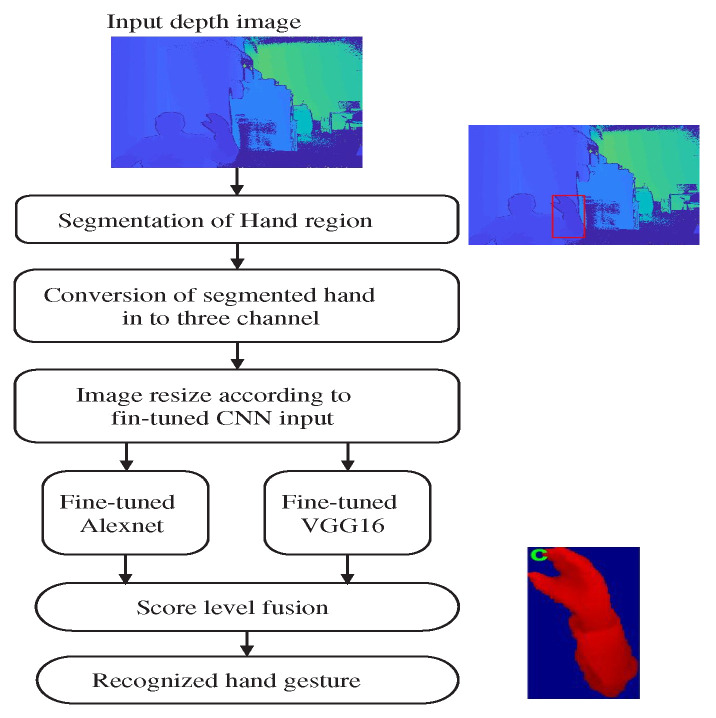
Flow chart for development of real-time gesture recognition system.

**Figure 9 sensors-22-00706-f009:**
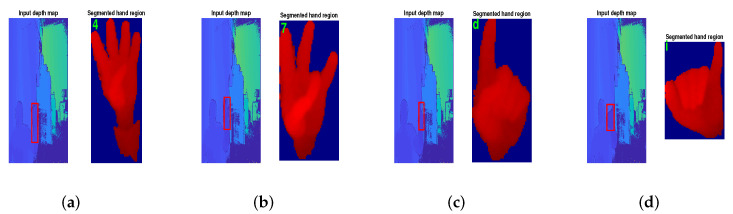
The hand region part is segmented from the input depth map, and the recognized gesture pose is displayed in the figure. (**a**–**d**) are different detected real-time gesture poses of ‘4’, ‘7’, ‘d’, and ‘i’ respectively using proposed method.

**Table 1 sensors-22-00706-t001:** Analysis of different sensors used for the recognition of hand gestures.

Data Acquisition Sensors	Wearable	Advantages	Limitations
Data glove	Yes	Low cost, robust	Less comfort and less user-friendly
Leap motion	No	Track hand with absolute precision	Always putting hand above the sensor and less coverage area
Vision sensors (Web camera)	No	Free to use	Affected by background and human noise
Depth sensor (Kinect)	No	No color marker, hand segmentation easier	Hand should be the first object in the camera frame

**Table 2 sensors-22-00706-t002:** Comparison results of mean accuracy with fine-tuned CNNs and score fusion for both CV tests on two standard datasets. Both the weight and bias of the pre-trained CNNs are fine-tuned for end-to-end layers.

Fine-Tuned CNN	MU Dataset		HUST-ASL Dataset	
	LOO CV, %	Regular CV, %	LOO CV, %	Regular CV, %
AlexNet	87.10 ± 1.67	97.88 ± 1.71	49.26 ± 3.66	61.05 ± 1.39
VGG-16	88.11 ± 1.44	97.80 ± 1.72	54.71 ± 3.34	62.51 ± 1.04
Proposed	90.26 ± 1.35	98.14 ± 1.68	56.18 ± 3.13	64.55 ± 0.99

**Table 3 sensors-22-00706-t003:** Comparison of proposed technique with earlier techniques using LOO CV test on MU dataset.

Test Methods	Mean Accuracy (LOO CV), %
CNN [24]	73.86 ± 1.04
FFCN [24]	84.02 ± 0.59
AlexNet + SVM [28]	70.00
VGG 16 + SVM [28]	70.00
Proposed	90.26 ± 1.35

**Table 4 sensors-22-00706-t004:** Comparison of proposed technique with earlier techniques using holdout CV test on MU dataset.

Test Methods	Mean Accuracy (Holdout CV), %
GB-ZM	97.09 ± 0.80
GB-HU	97.63 ± 0.76
Proposed	98.93 ± 0.68

**Table 5 sensors-22-00706-t005:** Comparison of proposed technique with earlier techniques in leave-one-subject-out CV test on HUST-ASL dataset.

Test Methods	LOO CV	Regular CV
Front-view-based BCF	50.4 ± 6.1	56.5 ± 0.6
Proposed	56.27 ± 3.13	64.51 ± 0.99

**Table 6 sensors-22-00706-t006:** Error analysis on ASL hand gesture datasets without any fingers held out.

ASL Gesture Class	MU Dataset, %			HUST-ASL Dataset, %		
	AlexNet	VGG-16	Proposed	AlexNet	VGG-16	Proposed
0	62.9	61.4	67.1	45.0	51.9	53.1
a	97.1	100	100	24.4	36.3	33.8
e	100	100	100	16.9	29.4	29.4
m	75.7	58.6	64.3	38.1	41.3	43.1
n	70.0	88.6	88.6	30.0	37.5	38.8
o	77.1	78.6	84.3	32.5	33.1	35.0
s	95.7	98.6	98.6	23.1	26.3	28.8
t	95.4	100	100	38.8	36.9	40.0

## Data Availability

Publicly available datasets are utilized in this work. The datasets are available at mu dataset: https://www.massey.ac.nz/~albarcza/gesture_dataset2012.html; HUST-ASL dataset: http://mc.eistar.net/UpLoadFiles/File/hust_asl_dataset.zip.

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
