# Peer review of "Real-Time Hand Gesture Recognition Using Fine-Tuned Convolutional Neural Network"

_sensors, 2022, doi:10.3390/s22030706_

Round 1

Reviewer 1 Report

The manuscript entitled “Real Time Hand Gesture Recognition using fine-tuned Convolutional Neural Network.” This paper aims to propose the use of a fine-tuning approach in a pre-trained convolutional neural network to recognize real-time hand gestures. This is a well-written paper and addresses an important topic for a broad readership of the Journal Sensors, given the posited challenges of background interference and accuracy in hand gesture recognition used in sign language and general knowledge on the usage implications of such models in real-time transfer learning.

Several assertions are too simplistic – e.g. “Hand gesture recognition is one of the most effective mode of interaction between human and computer due to higher flexibility and user friendly.” Information relative to the way it is effective in comparison with other modes of interaction is missing. Please explain in what way it is flexible and user-friendly.

Lack of clarity in terms of the scope of the paper and interrelate with the gap in the literature and advances in the field need to be resolved. References are applicable but some are missing to support several assertions that would need evidence/support. e.g. ‘the static hand gestures provide higher flexibility and are user friendly.’ The following publication can also help to sustain some considerations about gestural interaction (both in development and usage): McNeill, D. (1992).Hand and mind: What gestures reveal about thought. Illinois, USA: University of Chicago Press.

Further information on data acquisition relative to the challenges and implications of the use of web cameras, Kinect-depth sensors, leap motion, smart color glove-based systems would be also valuable to justify the study rationale. Then, lack of organization when describing the undertaken procedures may compromise readership and prevent the authors to make their point clear – I’d recommend organizing the method section by starting with the data acquisition process, and then advancing with the segmentation of hand region, feature extraction, and gesture classification.

I’d also recommend the authors illustrate the transfer learning process, emphasizing the source and target domains to improve readership and understanding of the proposed model.

The authors could also strengthen the conclusion section by providing some minor implications from their findings, knowledge contributions in pre-trained CNN, and future work.

Minor editing:

Real Time > Real-time

Line 1 – one of the most effective mode of interaction > one of the most effective modes of interaction

Cross validation > cross-validation

Involves most > involves the most

User friendly > user-friendly

In this cases > In these cases

Classification steps fails > classification steps fail

To trained the CNN from > to train the CNN from…

Show better result > show better results than

The detail experimental results > The detailed experimental results

Line 89: To over come this difficulty  > to overcome this difficulty

Ren et al. [6], proposes a novel feature > Ren et al. [6], propose a novel feature

contour features is limited > contour features are limited

is applied to recognized the gesture >  is applied to recognize the gesture

mimimum depth > minimum depth

The distinguish features are extracted > The distinguishing features are…

hight 165 > height 165

… among others. I recommend the authors to make thorough proofreading.

Author Response

Reviewer 1 comments and authors’ response

1. Several assertions are too simplistic – e.g. “Hand gesture recognition is one of the most effective mode of interaction between human and computer due to higher flexibility and user friendly.” Information relative to the way it is effective in comparison with other modes of interaction is missing. Please explain in what way it is flexible and user-friendly.

Authors’ Response: Hand gesture recognition is more flexible and user friendly compared to remote control and joystick because in vision-based hand gesture interface system, the user has to operate the machine using his hand in front of the camera. Also the user need not to wear any wearable device on his hand which makes it uncomfortable and feels free to operate it. As recommended by the reviewer, the above explanation is included in the ‘Introduction’ section of the revised manuscript. (Page no. 1)

2. Lack of clarity in terms of the scope of the paper and interrelate with the gap in the literature and advances in the field need to be resolved. References are applicable but some are missing to support several assertions that would need evidence/support. e.g. ‘the static hand gestures provide higher flexibility and are user friendly.’ The following publication can also help to sustain some considerations about gestural interaction (both in development and usage): McNeill, D. (1992). Hand and mind: What gestures reveal about thought. Illinois, USA: University of Chicago Press.

Authors’ Response: As suggested by the reviewer, the recent works on vision-based hand gesture recognition technique is included in the revised manuscript which improve the scope of the paper. As recommended by the reviewer, some recent techniques available in this field are added in both ‘Introduction’ and ‘related works’ section of the revised manuscript. (Page no.s 1 and 2). Also, the suggested paper is cited in the revised manuscript.

3. Further information on data acquisition relative to the challenges and implications of the use of web cameras, Kinect-depth sensors, leap motion, smart color glove-based systems would be also valuable to justify the study rationale. Then, lack of organization when describing the undertaken procedures may compromise readership and prevent the authors to make their point clear – I’d recommend organizing the method section by starting with the data acquisition process, and then advancing with the segmentation of hand region, feature extraction, and gesture classification. I’d also recommend the authors illustrate the transfer learning process, emphasizing the source and target domains to improve readership and understanding of the proposed model.

Authors’ Response: As suggested by the reviewer, the advantages and limitations of different data acquisition sensors such as dat glove, leap motion, web cameras, and Kinect-depth sensors are discussed in the new ‘Data acquisition’ section of the revised manuscript (page no. 4 ).

The revised manuscript is also well organized according to the reviewer suggestion. The methodology of the revised manuscript is also started with the ‘Data acquisition’ section and the next sections are arranged accordingly.

As suggested by the reviewer, the transfer learning process is visualized in the Figure. 3 and discussed in the revised manuscript (Page no. )

4. The authors could also strengthen the conclusion section by providing some minor implications from their findings, knowledge contributions in pre-trained CNN, and future work.

Authors’ Response: As suggested by the reviewers the ‘conclusions’ section of the revised manuscript is modified by adding the limitation and future scope of the proposed work.

5. Minor editing: Real Time > Real-time Line 1 – one of the most effective mode of interaction > one of the most effective modes of interaction Cross validation > cross-validation Involves most > involves the most User friendly > user-friendly In this cases > In these cases Classification steps fails > classification steps fail To trained the CNN from > to train the CNN from. . . Show better result > show better results than The detail experimental results > The detailed experimental results Line 89: To over come this difficulty > to overcome this difficulty Ren et al. [6], proposes a novel feature > Ren et al. [6], propose a novel feature contour features is limited > contour features are limited is applied to recognized the gesture > is applied to recognize the gesture mimimum depth > minimum depth The distinguish features are extracted > The distinguishing features are. . . hight 165 > height 165 . . . among others. I recommend the authors to make thorough proofreading.

Authors’ Response: As suggested by the reviewers the minor corrections are rectified in the revised manuscript.

Reviewer 2 Report

This research suggested an end-to-end fine tuning method of a pre-trained CNN model using score level fusion technique. The recognition performance of hand gesture was improved by the fusion of different deep CNN models. In general, this manuscript was well written and understood. However, the following questions and suggestions are presented to improve it.

1. You should explain why you selected two models, AlexNet and VGG-16, among various CNN models proposed in the previous studies. 
2. More detailed explanations regarding ‘Normalized score’ and ‘Score level fusion’ would be beneficial to the readers. The proposed model seemed to be developed using ‘Normalized score’ and ‘Score level fusion’, but the application of ‘Normalized score’ and ‘Score level fusion’ was not sufficiently explained. (3.3. Normalization & 3.4. Score level fusion technique between two fine tuned CNNs)
3. Describe the MU dataset in more detail, such as performed gestures, repetition times, and reasons of selection in this study, etc. (4.1.1. Massey University (MU) dataset)
4. The sentence was represented by “Precision of system accuracy”. Isn’t it just “precision”? (line 24)
5. The program and the toolbox for deep learning were not mentioned. Please explain them.
6. The pre-trained model seems to be developed using the transfer learning, but there was no explanation which dataset was used for the transfer learning. In my opinion, the pre-trained model would not be effective when the type of training image was different from that of target image. (3.2 Archtecture of pre-trained CNNs and fine-tuning)
7. In general, the classification performance relates with the amount of training data. However, Regular CV showed better classification performance than LOO CV, even though LOO CV had larger training data than Regular CV. You should explain this results in more detail. (Table 1)
8.  Holdout CV seems to have the same meaning as a Regular CV. If they have same meaning, terminology should be unified. If they have different meaning, an additional explanation would be necessary. (Table 3) 
9. The results of Real-time recognition were not represented. You have to represent the results (latency and accuracy) of Real-time system. (6. Recognition of ASL gesture in real-time)

Author Response

Reviewer 2 comments and authors’ response

1. You should explain why you selected two models, AlexNet and VGG-16, among various CNN models proposed in the previous studies.

Authors’ Response: Authors initially experimented with the different CNN architectures, but those are not suited to the specifically chosen gesture recognition. Also, from the various literature, the remaining CNN architectures are not crossing more than 80% of accuracy in gesture recognition. After several attempts, it is observed that AlexNet and VGG-16 are providing the best performance in gesture recognition. VGG-16 will be more efficient in size and training time (because there are fewer layers than any pre-trained model trained for more than one purpose). The following advantages make them better in proposed gesture recognition (i) VGG-16 had 13 convolution layers, which extracts in-depth features from the input image, (ii) AlexNet allows for multi-GPU training by putting half of the model’s neurons on one GPU and the other half on another GPU, which helps in training of bigger models, but it also cuts down on the training time.

2. More detailed explanations regarding ‘Normalized score’ and ‘Score level fusion’ would be beneficial to the readers. The proposed model seemed to be developed using ‘Normalized score’ and ‘Score level fusion’, but the application of ‘Normalized score‘ and ‘Score level fusion’ was not sufficiently explained. (3.3. Normalization and 3.4. Score level fusion technique between two fine-tuned CNNs)

Authors’ Response: As suggested by the reviewer, the detailed explanations regarding ‘Normalized score’ and ‘Score level fusion’ are included in the revised manuscript os the subsection ‘Normalization’ and ‘Score level fusion technique between two fine-tuned CNNs’ respectively.

3. Describe the MU dataset in more detail, such as performed gestures, repetition times, and reasons of selection in this study, etc.

(4.1.1. Massey University (MU) dataset)

Authors’ Response: As suggested by the reviewer, the detail explanation regarding the dataset is included in the revised manuscript of the subsection ‘Massey University (MU) dataset. Since in this work, our objective is to recognize the ASL gesture poses that’s why we used this dataset to test the gesture recognition performance using our proposed technique.

4. The sentence was represented by “Precision of system accuracy”. Isn’t it just “precision”? (line 24)

Authors’ Response: Yes, it is corrected in the revised manuscript.

5. The program and the toolbox for deep learning were not mentioned. Please explain them.

Authors’ Response: In this work the Matlab deep learning toolbox is used to develop the proposed system. As suggested by the reviewer, it is included in the revised manuscript in the subsection ‘setting of hyper-parameters for fine-tunning’.

6. The pre-trained model seems to be developed using the transfer learning, but there was no explanation which dataset was used for the transfer learning. In my opinion, the pre-trained model would not be effective when the type of training image was different from that of target image. (3.2 Architecture of pre-trained CNNs and fine-tuning)

Authors’ Response: The pre-trained model is generally fine-tuned using the target dataset. Both the pre-trained AlexNet and VGG 16 nets are fine-tuned using the MU dataset training gesture samples and the performance is evaluated on the testing samples using the proposed technique. Similarly, for the HUST-ASL dataset both the pre-trained CNNs are fine-tuned on using the training gesture samples of the HUST-ASL dataset and tested on the test images of the HUST-ASL dataset. Again HUST-ASL training model is used to recognize the ASL gestures with the RGB-D sensor in real-time.

7. In general, the classification performance relates with the amount of training data. However, Regular CV showed better classification performance than LOO CV, even though LOO CV had larger training data than Regular CV. You should explain this results in more detail. (Table 1)

Authors’ Response: The LOO CV technique is a user-independent CV technique [9]. In this technique, the performance of the

trained model is evaluated using the gesture samples of the user who does not take part in the model development. However, in regular CV, the gesture samples of all the users in the dataset take part in the training and testing process. Hence this CV test is user-biased. Therefore, the model performance using the regular CV test is higher than the LOO CV test. The above explanation is also added in the revised manuscript of the subsection ‘performance evaluation’.

8. Holdout CV seems to have the same meaning as a Regular CV. If they have same meaning, terminology should be unified. If they have different meaning, an additional explanation would be necessary. (Table 3)

Authors’ Response: Hold out CV and regular CV techniques are different in sense of preparing the training and testing data. In the Hold out CV test, 80% gesture samples of the dataset are used to train the model, and the rest 20% samples are used to test the model performance [24]. In regular CV tests, the splitting ratio training and testing samples are 50% Vs 50% respectively [9]. The clarification regarding this is also included in the revised manuscript of the subsection ‘Comparison results on MU dataset’.

9. The results of Real-time recognition were not represented. You have to represent the results (latency and accuracy) of Real-time system. (6. Recognition of ASL gesture in real-time)

Authors’ Response: The recognition of ASL

Reviewer 3 Report

The work summary:

The authors addresses  the static hand gesture recognition problem using two pre-trained CNNs. The pre-trained CNNs are fine tuned on the target datasets. The output score of the two fine tuned CNNs are combined together to recognized the proper gesture class. 

Due to unavailability of large labeled image samples in static hand gesture images, transfer learning technique is adopted in image classification for small image sample datasets.

Typos:

Page 3 line 104 (senors ?)

Page 5 line 194 in state of training …. (instead of ?)

Page 11 - Line 25 (sore ?)

Questions to authors:

1) Static hand gestures can be properly applied to Signal language recognition ? Many "words" in this language requer a combination of more than one gesture, therefore, image sequence is needed to deal with recognition for words. Thus, the proposed work has limited application in real life applications for Signal language.

2) Confusion among 0, a, e, m, n, o, s, t symbols may come from that the depth method is unable to distinguish small depth variation ? How to overcome that ? Have you check if including image features such as edge information could improve the performance for these cases ?

3) As the goal is real time processing why the authors used Matlab instead of OpenCV ? They mention the use of NVIDIA Card, it can be used properly for improving performance with Matlab ?

Author Response

Reviewer 3 comments and authors’ response

1) Static hand gestures can be properly applied to Signal language recognition? Many "words" in this language require a combination of more than one gesture, therefore, image sequence is needed to deal with recognition for words. Thus, the proposed work has limited application in real life applications for Signal language.

Authors’ Response: We agree with the reviewer’s statement but our technique mainly focuses on American sign language (ASL) recognition as it is operated on a single hand. So, we have not applied the technique using the sign language dataset with both hands. In future work, we may focus on the same.

2) Confusion among 0, a, e, m, n, o, s, t symbols may come from that the depth method is unable to distinguish small depth variation? How to overcome that? Have you check if including image features such as edge information could improve the performance for these cases?

Authors’ Response: We agree with the reviewer’s statement that there is confusion among similar gestures using depth images. To overcome that, some image features from the segmented hand gesture can be incorporated with the proposed HGR system in feature fusion mode, which may improve the performance. Though we have not checked the same, it will be our future direction of the research study.

3) As the goal is real time processing why the authors used Matlab instead of OpenCV? They mention the use of NVIDIA Card; it can be used properly for improving performance with Matlab?

Authors’ Response: Authors are expertise in both MATLAB and Open CV. But authors have faced difficulties with version control dependency in Open CV while executing the algorithms. Therefore, algorithm development, modelling, simulation, and prototyping are implemented using MATLAB to generate GPU-based codes that run directly on GPU processors with minimal engineering efforts to get improved performance.

Round 2

Reviewer 2 Report

This manuscript was well revised, but the results of Real-time recognition were still not represented.
Results such as latency or classification accuracy should be presented or discussed, as mentioned in the title of the manuscript “Real time hand gesture recognition using fine-tuned convolutional neural network”.

Author Response

1. Results such as latency or classification accuracy should be presented or discussed.

Authors’ Response: As suggested by the reviewers, the computational time (s) to recognized a gesture pose using the proposed method is included the revised manuscript. The pre-processing time of the proposed system for hand region segmentation with image resizing of the input depth image is 0.0969 s. The fine-tuned CNNs utilized 0.4236 s to generate individual scores, and finally, recognition of gesture using the score fusion technique takes 0.0014 s. Thus, the total time to recognize a gesture pose using the proposed technique is 0.5219 s.

The above explanation is also added in the revised manuscript of the subsection ‘Computational time of the proposed method on real-time data’ (Page no. 11).
